# The Supplementary Motor Area Responsible for Word Retrieval Decline After Acute Thalamic Stroke Revealed by Coupled SPECT and Near-Infrared Spectroscopy

**DOI:** 10.3390/brainsci10040247

**Published:** 2020-04-22

**Authors:** Shigeru Obayashi

**Affiliations:** 1Department of Rehabilitation Medicine, Dokkyo Medical University Saitama Medical Center, 2-1-50 Minami-Koshigaya, Koshigaya, Saitama 343-8555, Japan; 2Department of Rehabilitation Medicine, Chiba-Hokusoh Hospital Nippon Medical School, 1715 Kamagari, Inzai, Chiba 270-1694, Japan

**Keywords:** cerebro-cerebellar diaschisis, FAT: frontal aslant tract, functional near-infrared spectroscopy (f-NIRS), hemodynamic response, perfusion, SPECT, thalamic aphasia, verbal fluency test

## Abstract

Damage to the thalamus may affect cognition and language, but the underlying mechanism remains unknown. In particular, it remains a riddle why thalamic aphasia occasionally occurs and then mostly recovers to some degree. To explore the mechanism of the affected cognition and language, we used two neuroimaging techniques—single-photon emission computed tomography (SPECT), suitable for viewing the affected brain distribution after acute thalamic stroke, and functional near-infrared spectroscopy (f-NIRS), focusing on hemodynamic responses of the supplementary motor area (SMA) responsible for speech production in conjunction with the frontal aslant tract (FAT) pathway. SPECT yielded common perfusion abnormalities not only in the fronto–parieto–cerebellar loop, but also in the SMA, IFG and surrounding language-relevant regions. In NIRS sessions during a phonemic verbal fluency task, we found significant word retrieval decline in acute thalamic patients relative to age-matched healthy volunteers. Further, NIRS showed strong correlation between word retrieval and posterior SMA responses. In addition, follow-up NIRS exhibited increased bilateral SMA responses linked to improving word retrieval ability. The findings suggest that cognitive dysfunction may be related to the fronto–parieto–cerebellar loop, while language dysfunction is attributed to the SMA, IFG and language-related brain areas. SMA may contribute to the recovery of word retrieval difficulty and aphasia after thalamic stroke.

## 1. Introduction

Increasing evidence has revealed that the thalamus plays a crucial role for cognitive function and language by divergent and convergent thalamocortical and corticothalamic pathways in a complementary manner [1]. In particular, thalamic aphasia may occasionally present after thalamic stroke and then mostly recover to a certain degree [2]. However, it remains a puzzle why aphasia takes place after damage to the thalamus and how patients can recover from aphasia.

The supplementary motor area (SMA) contributes to speech production [3,4] as well as motor control and executive function [5,6]. SMA was subdivided into two domains: the SMA proper and pre-SMA. Further, SMA may be involved in verbal fluency via the frontal aslant tract (FAT) [7,8]. Damage to pre-SMA and SMA can also lead to motor and speech deficits. However, in some cases, the disorders are only temporary, resolving within weeks to months [9,10]. A previous report suggested SMA involvement in the early phase of the language recovery process after stroke [11]. In addition, we found that SMA may play a crucial role in aging-related word retrieval difficulties using near-infrared spectroscopy (NIRS) [12].

The ventral lateral (VL) nucleus of the thalamus projects mainly to area 6 including the SMA proper, while the mediodorsal (MD) nucleus projects to area 8 including pre-SMA, and the medial geniculate (MG) nucleus projects to areas 41 and 42 (superior temporal gyrus (STG)), mostly related to language processing. Given the heterogeneity of thalamic nuclei in terms of function and projections to different areas of the cortex, it is also likely that damage to the thalamus may generate a variety of cognitive impairments.

So far, little attention has been paid to cognitive declines caused by subcortical damage. A possible mechanism could be attributable to cerebro-cerebellar diaschisis [13], which might be related to the fronto–cerebellar–thalamic loop via the brainstem. In fact, brainstem stroke produced cognitive dysfunction, probably as a result of the relation to frontal hypo-activity [14,15,16,17]. We recently investigated the effect of brainstem lesions on SMA activity via the fronto–cerebellar–thalamic loop using coupled SPECT and functional near-infrared spectroscopy (f-NIRS) with a phonemic verbal fluency task (VFT) [17]. Our results suggest that SMA activity may predict a positive prognosis of cognitive dysfunction after pontine infarct. Together, we propose two hypotheses. One is that cognitive impairment due to acute thalamic stroke may be related to the fronto–brainstem–cerebellar–thalamic loop, similar to the system underlying cognitive dysfunction due to acute pontine stroke. The other is that language dysfunction due to acute thalamic stroke involves the language-related brain regions including the SMA, inferior frontal gyrus (IFG), STG, angular gyrus and supramarginal gyrus.

The present study used both SPECT and f-NIRS to address (1) whether thalamic stroke generates cognitive impairment and language dysfunction, (2) the distance between and expansion of the affected brain areas due to thalamic stroke, (3) whether SMA activity is affected by acute thalamic stroke and whether the changes are associated with cognitive and language function, and (4) whether such associations are related to recovery of the dysfunction. 

## 2. Materials and Methods

### 2.1. Participant Recruitment and Inclusion Criteria

Acute patients with thalamic stroke were recruited at the acute rehabilitation unit of Chiba-Hokusoh Hospital, Nippon Medical School, between June 2010 and June 2012. In Japan, the duration of acute rehabilitation for a stroke is set at 2 to 3 weeks after onset. All 39 patients had an ischemic stroke involving the thalamus. Inclusion criteria for neuroimaging included patients under age 80, with first-ever isolated thalamic infarcts or hemorrhages without motor deficits. Exclusion criteria were a past or current history of psychiatric or neurological illnesses as well as patients with motor impairment due to the ictus. Finally, 27 patients met the criteria, so a total of 27 acute patients with thalamic stroke (including 15 with hemorrhage; affected hemisphere: right, left, bilateral = 18, 7, 1; average age 63.3 ± 6.17 years; Glasgow Coma Scale score: 13–15) were recruited (Figure 1). Sixteen patients underwent a series of SPECT studies with ^99m^Tc-ECD. Further, a total of 28 subjects, including 17 patients with thalamic stroke (9 males and 8 females, average age 63.3 ± 6.17 years) and 11 age-matched normal healthy subjects (3 males and 8 females, average age 63.3 ± 7.39 years), participated in f-NIRS with VFT. Neuroimaging measurements and neuropsychological tests were conducted within one week of onset. The subjects provided written informed consent after detailed explanation of the procedures. This study was reviewed and approved by the Ethics Committee of Chiba Hokusoh Hospital, Nippon Medical School (Registered number: 228), and all procedures involved were concordant with the latest version of the Declaration of Helsinki.

### 2.2. Neuropsychological Estimation

All patients were assessed by well-trained neuropsychologists using the standardized Japanese translation of the mini-mental state estimation (MMSE) and a neuropsychological test battery consisting of the following tests: clinical assessment for attention (CAT), behavioral inattention test (BIT), and trail making test part A (TMT-A) for attention and processing speed, TMT-B (Japanese version, where Kana letters replaced the Roman alphabet), frontal assessment battery (FAB) and behavioral assessment of dysexecutive syndrome (BADS) for executive function, standard verbal paired-associate learning test (S-PA), Rivermead behavioral memory test (RBMT), Rey–Osterrieth complex figure test (ROCFT), and Wechsler memory scale test (WMS) for memory. The results of TMT were determined according to Toyokura’s report with Japanese subjects [18]. Further, the standard language test of aphasia (SLTA) was used to estimate aphasia.

### 2.3. NIRS Studies: Image Acquisition and Analysis

#### 2.3.1. Image Acquisition 

We measured dynamic changes in oxygenated hemoglobin concentration ([oxy-Hb]), deoxygenated hemoglobin concentration ([deoxy-Hb]), and total hemoglobin concentration ([total-Hb]) during phonemic VFT using the 22-channel NIRS system (ETG-4000 Optical Topography System; Hitachi Medical Co., Tokyo, Japan). The probes were positioned in accordance with the international 10/20 system for EEG electrode placement. The system utilizes two different wavelengths of near-infrared light (695 and 830 nm) with a temporal resolution of 10 Hz to measure its relative changes. These changes were transformed into the concentration changes of [oxy-Hb] and [deoxy-Hb] as indicators for brain activity by means of a modified Beer–Lambert law [19]. The distance between pairs of emission and detector probes was set at 3.0 cm, resulting in the detection of cerebral blood volume within 2–3 cm from the skin beneath each pair. We used a probe set (plastic panel) of optodes. The set consisted of 8 light emitters and 7 detectors, which resulted in 22 channels with an area of 12 × 6 cm on the scalp. The 3 × 5 probes were attached to each subject’s superior frontal area, whose long axis was arranged orthogonal to the line connecting inion to nasion. The probes were symmetrically positioned centering on Fz, in accordance with the international 10/20 system for EEG electrode placement. 

#### 2.3.2. Definition of Regions of Interest

In regard to the placement of regions of interest (ROIs), we focused on the medial frontal area around Fz, corresponding to the supplementary motor area. For analyses of brain activity, with reference to a previous article [20], we placed four ROIs on the SMA—left and right anterior ROIs located anteriorly to Fz and left and right posterior ROIs located posteriorly to Fz. They demonstrated the spatial relationship between the international 10/20 system for EEG electrode placement and normalized brain coordinates, such as the standard Montreal Neurological Institute (MNI) and Talairach stereotactic coordinates. Fz was positioned at x = −0.1, y = 54.9, z = 62.3 in Talairach coordinates, corresponding to Brodmann 8, namely the pre-SMA, and Cz at 0.6, −7.6, 93.8, i.e., Brodmann 6. At the least, the posterior portion of the optodes may well involve the SMA proper. 

#### 2.3.3. Task Procedure of f-NIRS Study 

Phonemic VFT has often been used for estimating the executive function of the frontal cortex, so it might be suitable for detecting any change in frontal activity. For details, see our previous paper [12]. In short, each session consisted of five phonemic VFT (20 s each run) and five control task (40 s each run) blocks. Throughout the NIRS measurement session, all subjects were forced to stay seated on their chair. In each 20 s VFT session, participants were required to produce as many nouns in Japanese as possible beginning with a certain letter without the use of repetitions. The control task consisted of repeating a train of syllables (“/a/, /i/, /u/, /e/, /o/”) at 1 Hz for 40 s for each run. The control condition followed the VFT condition in a counterbalanced manner.

#### 2.3.4. NIRS Data Acquisition and Analysis

NIRS data were sampled every 0.1 s. Basically, NIRS data include task-unrelated slow drifts in the measurement, so irrelevant components should be excluded before the analysis. The “integral mode” was the first-order correction to exclude irrelevant components during VFT, the process of which consists of linear fitting of baseline and averaging. Initially, the obtained data were automatically preprocessed using the “integral mode”, a command defined by the HITACHI NIRS machine. The pre-task baseline was determined as the mean across the last 5 s of the pre-task period, and the post-task baseline was determined as the mean across the last 5 s of a 25 s period after stimulation (VFT). Then, linear fitting was performed based on the data between the two baselines. Moving average methods were applied to remove short-term motion artifacts in the analyzed data (moving average window: 5 s). Following this correction, the segments of each condition were averaged. For analyses of brain activity, we placed 4 ROIs and defined two time segments, i.e., 5 s of the corrected baseline before the beginning of activation (VFT) and the last 10 s of stimulation, and then the mean values ([oxy-Hb], [deoxy-Hb], and [total-Hb]) of these segments were calculated. 

### 2.4. SPECT Image Acquisition and Analysis 

Thalamic stroke patients underwent a series of SPECT studies with ^99m^Tc-ECD. Brain perfusion images were obtained with a triple-head Philips IRIX SPECT scanner with high-resolution collimators (spatial resolution, 3.6 mm FWHM). SPECT scan was started 10 min after administering ^99m^Tc-ethylcysteinate diethylester (^99m^Tc-ECD) (FUJIFILM RI Pharma Co., Tokyo, Japan) with a bolus injection of 600 MBq, and data were collected for 24 min. The images were reconstructed by the filtered back projection (FBP) method using a Butterworth filter. The reconstructed images were corrected for gamma ray attenuation using Chang’s method. We used the Patlak Plot method to identify the localization of abnormal regional cerebral blood flow (rCBF) at rest and then performed analysis using the easy Z-score Imaging System (eZIS) version 4.0 [21]. Each SPECT image of the patients was anatomically standardized using SPM2 with an original ^99m^Tc-ECD template. The standardized images were then compared with the mean and SD of SPECT images of healthy control subjects using voxel-by-voxel Z score analysis after voxel normalization to global mean or cerebellar values: Z score = ([control mean] - [individual value])/(control SD). The SPECT image data set of healthy control volunteers was derived from the normal database of the e-ZIS program at the National Center Hospital for Mental, Nervous and Muscular Disorders, National Center of Neurology and Psychiatry, Tokyo, Japan. It comprised a total of 110 healthy volunteers. Following acquisition of the eZIS brain map, we used the voxel-based analysis—Stereotactic Extraction Estimation (vbSEE) program to automatically convert the acquired data into Talairach brain coordinating space [22]. Subsequently, we calculated the total of the coordinate data with the Z-value exceeding the threshold of the Z-value set as a significant finding (±2.0 SD). 

### 2.5. Statistical Analysis

The numbers of word retrieval and [oxy-Hb] during f-NIRS were compared between patients with thalamic stroke and the control group. Statistical analysis was performed by Mann–Whitney U test for non-parametric data using SPSS ver. 26. Results were accepted as statistically significant at P<0.05. In addition, Spearman’s rank-order correlation (r_s_) was run to determine the association between word retrieval and [oxy-Hb] and between SPECT Z scores and neuropsychological scores.

## 3. Results

### 3.1. Neuropsychological Findings

The averaged MMSE score was 25.87 (SD 2.75) for thalamic stroke survivors. Overall, neuropsychological results revealed that 25 of the 27 patients with acute thalamic stroke (92.6%) had affected cognition including inattention (18 patients), memory disturbance (15), executive dysfunction (11) and social behavioral disturbance (1). The results demonstrated disturbances in one cognitive domain for 9 patients, two domains for 12, and three domains for 4 (Table 1). Three patients presented with dysphasia as shown by SLTA (Table 2). 

### 3.2. SPECT Results 

#### 3.2.1. SPECT Z-Score Mapping 

SPECT data were obtained from 16 patients, yielding perfusion abnormality in distributed cortical brain areas including the frontal cortices, superior temporal cortices, parietal cortices and cerebellum. Table 3 shows the Z-scores of each brain area. In particular, damage to the thalamus commonly affected brain perfusion in language-related cortical areas including in Broca’s area (left 44, 45), Wernicke’s area (22, 42), angular gyrus (39), and supramarginal gyrus (40). Further, perfusion abnormality involved bilateral SMA (6, 8) and IFG (44 and 45), connected by the FAT pathway as well as bilateral basal ganglia connecting with SMA. With respect to thalamic aphasia, two patients (TH12 and 16) presenting with global aphasia, commonly yielded hypo-perfusion in Brodmann areas 8, 44 and 45 connected by the FAT, and other language-related brain areas in the left hemisphere (22 and 42, 39, 40) and hyper-perfusion in right language homologous areas (22 and 42, 40) (Figure 2b,c). One patient (TH16) revealed hyper-perfusion at the bilateral SMA proper (area 6), while the other (TH12) showed bilateral hypo-perfusion. Both patients recovered from aphasia 3 Mo after ictus. A patient (TH9) with persistent motor aphasia 1 year later yielded hyper-perfusion in SMA, IFG and other language-relevant areas in the left hemisphere without abnormality in right homologous areas (Figure 2a). The correlation of Z-scores with any neuropsychological ones were not significant.

#### 3.2.2. Follow-Up SPECT

Follow-up SPECT was conducted in only one patient (TH6: Figure 3). Perfusion abnormality, obtained from the second SPECT study, showed a different pattern from the first SPECT study; in particular, hyper-perfusion of left IFG returned to a normal level at the second SPECT, as word retrieval disability improved. Further, the right SMA consistently showed hyper-perfusion, whereas the left SMA perfusion changed in a reversed manner (Figure 3d). 

### 3.3. NIRS Results

#### 3.3.1. Word Retrieval: Comparison between Thalamic Stroke Survivors and Control

During NIRS sessions, the average number of word retrieval was 13.18 (SD: 6.4) for 16 patients with thalamic stroke, while that for the control group was 21.9 (SD: 5.375), showing a significant difference between the two groups (Mann–Whitney U = 25, Z = 3.114, *p* = 0.001) (Figure 4a and Table 4). The findings suggested that thalamic stroke survivors were characterized by word retrieval difficulty, regardless of almost absence of aphasia, as shown in Table 2.

#### 3.3.2. NIRS Results: The Comparison with the Control and the Correlation

To exclude irrelevant processing from VFT-derived brain activity, very low performance (only 0–3 words in total sessions, due to dysphasia or mutism; see Table 4) during NIRS measurements was discarded from analysis. As a result, NIRS data from 25 subjects (14 patients and 11 healthy subjects) were available for analysis. NIRS with phonemic VFT showed no significant difference between SFX [oxy-Hb] increases at anterior ROIs and those from the age-matched healthy control group, while SFX [oxy-Hb] increases at posterior ROIs were lower compared to control. Overall, there were no significant differences at either ROIs between the two groups. However, when patients were classified into three groups (retrieved words below -1.50 SD units from the mean of control group as poor performance group; -1.50 SD to the mean as moderate one; above the mean as good one) by severity of word retrieval decline compared to patients in the healthy control group, we confirmed a trend toward the association of [oxy-Hb] derived from NIRS and word retrieval ability (Figure 4b). In particular, the poor and moderate groups showed remarkably low SMA responses relative to the healthy group.

The numbers of word retrieval during the NIRS sessions had strong correlation with [oxy-Hb] increases during phonemic VFT at left posterior SMA ROI (r_s_ = 0.748, *p* = 0.002) and right posterior SMA ROI (r_s_ = 0.786, *p* = 0.001), but weak correlation with those at left anterior SMA ROI (r_s_ = 0.346, *p* = 0.206) and no correlation with those at right anterior SMA ROI (r_s_ = 0.156, *p* = 0.579) (Figure 4c).

#### 3.3.3. Follow-Up NIRS Tests

Three patients (TH6, 11, 14) were available for follow-up NIRS data. NIRS was performed three times for TH6, and two times for TH11 and TH14, respectively. All patients had improved word retrieval ability a few months after onset. Simultaneously, increased [oxy-Hb] at bilateral posterior SMA ROIs was also linked to improved VFT performance (Figure 5).

#### 3.3.4. NIRS Data and SPECT Z-Scores

When patients were classified into three groups by severity of word retrieval decline, some of the language-related brain regions were sensitive to the classification. Specifically, higher Z-scores at left posterior SMA, IFG and left STG were observed when word retrieval performance was better, while lower Z scores at right posterior SMA showed better performance (Figure 6). On the contrary, Z-scores at bilateral area 8 (b) and bilateral basal ganglia (c) were seemingly independent of word retrieval disability.

## 4. Discussion

The present study demonstrated that thalamic stroke produced cognitive dysfunction. Further, thalamic stroke significantly affected language function, particularly word retrieval, and posterior SFX responses had strong correlation with word retrieval. Furthermore, improvement in word retrieval decline over time was linked with increased SFX activities. SPECT studies yielded perfusion abnormality involving SMA and IFG via FAT and language-related brain areas as well as the frontal-parieto–cerebellar loop.

The thalamus projects to all areas of the cortex, including those in the frontal, temporal, and parietal cortical regions associated with cognition and language. The ventral lateral (VL) nucleus of the thalamus projects mainly to area 6 including the SMA proper, while the mediodorsal (MD) nucleus projects to area 8 including pre-SMA, and the medial geniculate (MG) nucleus projects to areas 41 and 42 (superior temporal gyrus). Given the heterogeneity of thalamic nuclei in terms of function and projections to different areas of the cortex, it is more likely that damage to the thalamus may generate a variety of cognitive impairments. Our data revealed that most thalamic stroke survivors had affected cognitive function, such as memory, attention, executive function and social behavior. Consistently, SPECT studies yielded cerebral perfusion abnormality in a diffuse manner, including the frontal cortex, parietal cortex, temporal cortex, and cerebellum. These findings were consistent with our recent report, showing frontal hyper-perfusion and cerebellar hypo-perfusion via the fronto–brainstem–cerebellar–thalamic loop as well as cognitive disability after pontine infarct [17]. Thus, the abnormality may be related to cognitive impairment.

Given that non-aphasic patients had good performance of naming (see Table 2), their word retrieval difficulty could not be due mainly to dysphasia. Phonemic VFT requiring speech production involves both language function and executive function. Although it is difficult to isolate them each other, at least, word retrieval of aphasic patients was more affected by aphasia than executive function, whereas word retrieval of non-aphasic patients was more affected by executive function. We collected NIRS data from 16 patients including aphasic patients. As expected, aphasic patients had very poor performance (only 0–3 words retrieved words during a total 100 s). In addition, they could not thoroughly repeat a train of syllables as a control task. Throughout the NIRS session, they seemed to be silent with confusion. As a result, their brain activity in such a condition may mainly reflect cognitive and affective components, other than word retrieval ability. Therefore, we excluded aphasic patients from NIRS analysis.

The frontal aslant tract (FAT) has been associated with verbal fluency performance. Pre-SMA and SMA connects with IFG via FAT [7]. The phonemic verbal fluency task recruits the left IFG [23] and pre-SMA/SMA [5,24]. Specifically, left FAT is associated with speech production, whereas right FAT is involved in executive function/inhibition control [25]. Both FAT pathways may be involved in sequential motor planning. Our NIRS results, showing the close association of SFX responses with word retrieval ability regardless of the damage laterality, suggested that phonemic VFT required both FAT pathways to achieve both executive and linguistic functions. In fact, the right hemisphere was affected in the majority of our enrolled patients. Accordingly, the possible neural mechanism underlying word retrieval disability may be the association with FAT. A previous report suggested SMA involvement in the early phase of the language recovery process post-stroke [11]. SPECT results yielding perfusion abnormality in FAT may strengthen this hypothesis. An earlier study suggested that the left SMA may be involved in improvement of aphasia by rTMS [26]. Similarly, our results suggest that the left SMA showed positive correlation, while the right SMA showed negative correlation. The role of SMA on word retrieval ability might be the laterality dependent and, at the very least, the left and the right SMA may play a crucial role on word retrieval in different way. Damage to pre-SMA/SMA can also lead to motor and speech deficits. However, in most cases, the disorders are only transient, resolving within weeks to months [9,10]. Consistently, follow-up NIRS studies have demonstrated that word retrieval ability improved to the age-matched healthy subject’s level and that the improvement was seemingly linked to posterior SFX responses. Together, it is suggested that SMA and IFG via FAT pathway may contribute to the recovery from word retrieval difficulty after thalamic stroke.

The damage to the thalamus, more frequently in the left side than the right side, may affect language function [26,27,28]. Another finding from our SPECT studies is that despite the absence of dysphasia as shown by SLTA, all of the patients demonstrated perfusion abnormality in language-relevant brain regions regardless of the stroke laterality. The findings may include the compensate mechanism to protect against dysphasia after thalamic stroke. Thalamic aphasia is characterized by a relatively rapid recovery from language dysfunction. Most patients recover to a significant degree within 6 months post-stroke [2], although aphasia persists in some of the patients [29,30]. Consistently, in our study, two patients recovered from global aphasia, but the third patient’s affected motor speech continued, although the word retrieval disability seemed to be improving. The difference between aphasia-recovered patients and aphasia-persistent patients may be that the former demonstrated more diffuse perfusion abnormality, including in left language-relevant brain areas and right homologous areas compared to the latter showing abnormality restricted to only left language areas. So far, homologous language areas in the right hemisphere were considered to play a compensatory role for the recovery process of aphasia after stroke [11,31,32]. Our findings are consistent with this view. Further, our SPECT results showed perfusion abnormality in bilateral basal ganglia, suggesting another association with word retrieval difficulty after acute thalamic stroke. This view is explained by the fact that SMA also connects with the basal ganglia [33]. Earlier papers argued the contribution of basal ganglia and SMA to recovery from aphasia [34,35].

### Limitations

The limitations of our study have to be acknowledged. First, the present results were derived from not only a small sample of patients but also a right-dominant population in the affected hemisphere. Further, only a few patients could conduct follow-up NIRS and SPECT. One unique characteristic of rehabilitation medicine in acute care hospitals in Japan is that it is concluded within a restricted time period of approximately 2 weeks and it is subsequently continued in proper restorative rehabilitation care units of other hospitals. Second, the physiological significance of hyper-perfusion is not yet clear, while hypo-perfusion in a given brain area reflects any ischemic event or low neuronal and/or metabolic activity. Third, the spatial detectability of NIRS is limited. In addition, NIRS signals may also contain a scalp-derived component as well as a gray matter-derived component [36]. In any event, to date, a number of methods for the isolation of actual brain activity have already been presented.

## 5. Conclusions

This is the first report to disclose word retrieval difficulty as well as cognitive impairments in acute thalamic stroke survivors. The difficulty may be associated with the SMA and IFG systems via the FAT pathways and the surrounding language-relevant brain areas. Further, SMA may play a crucial role in improvement of word retrieval difficulty. A better understanding of the neurophysiological underpinnings of cognitive and language dysfunction after thalamic stroke could be expected to contribute to the establishment of therapy for the deficits.

## Figures and Tables

**Figure 1 brainsci-10-00247-f001:**
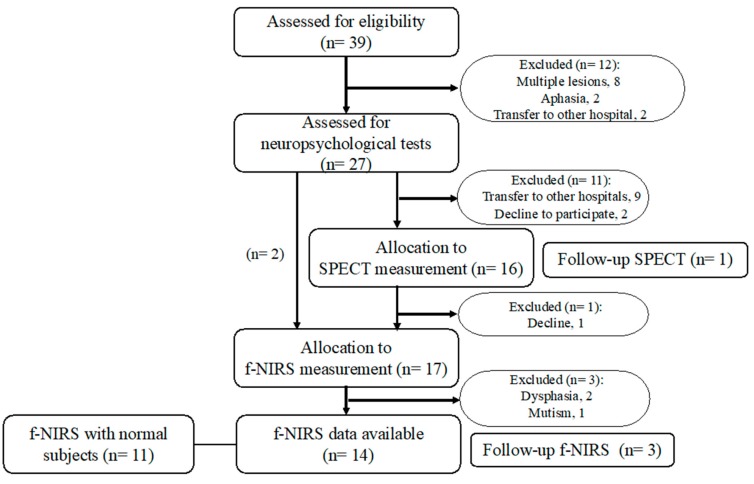
Flowchart showing the study selection process. Note that two patients declined to participate in single-photon emission computed tomography (SPECT), but participated in a NIRS session.

**Figure 2 brainsci-10-00247-f002:**
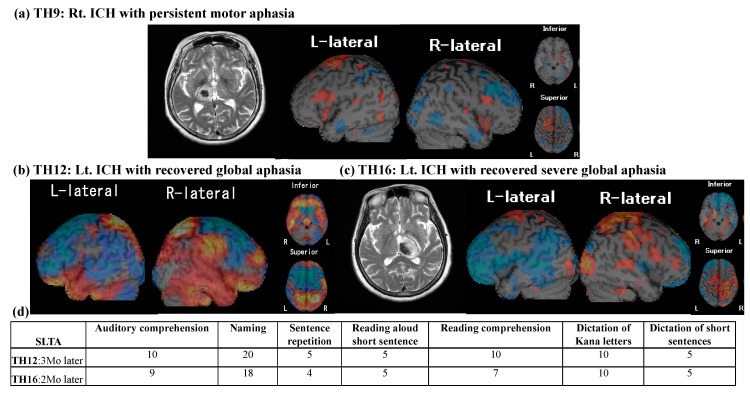
Initial SPECT results of patients with aphasia. (**a**) Axial view of T_2_-weighted MRI imaging and SPECT Z-map (from patient TH9). TH9 suffered from motor dysphasia due to right thalamic hemorrhage, and the dysphasia persisted for one year. L: left; R: right. (**b**,**c**) Axial view of T_2_-weighted MRI imaging and SPECT Z-maps (from patients TH12 and TH16). (**d**) Follow-up SLTA. Both patients suffered from global aphasia after left thalamic hemorrhage, but they recovered from aphasia within a few months after onset. Mo: month.

**Figure 3 brainsci-10-00247-f003:**
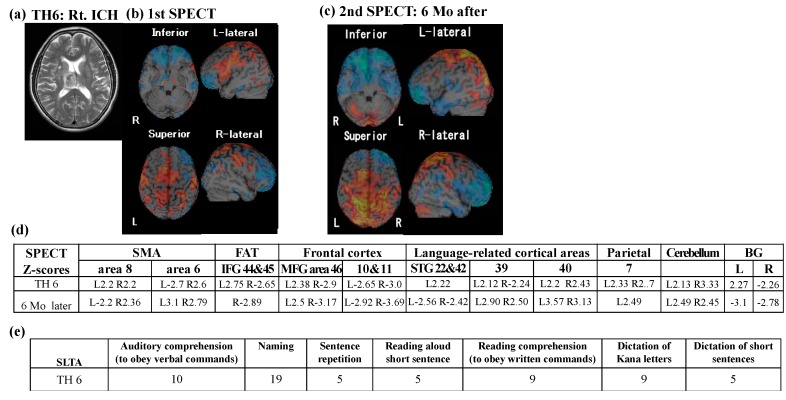
Repeated SPECT results from TH6. (**a**) Axial view of T_2_-weighted MRI. (**b**) First SPECT Z-map. L: left; R: right. (**c**) Second SPECT Z-map 6 months later. (**d**) Details of regional SPECT Z scores. Note that hyper-perfusion in left inferior frontal gyrus (IFG) returned to a normal level 6 months later. (**e**) SLTA results at first SPECT. Mo: month.

**Figure 4 brainsci-10-00247-f004:**
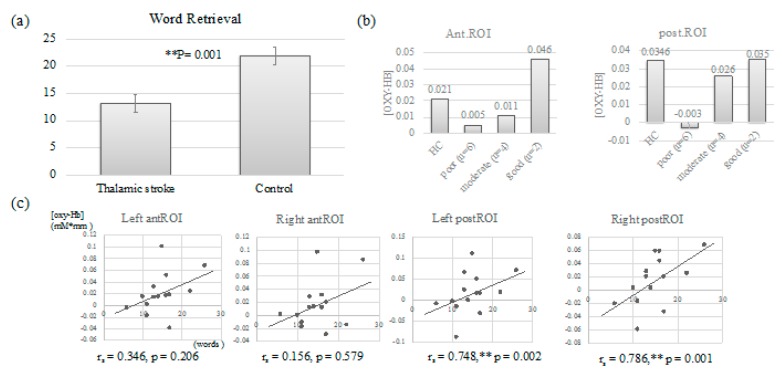
(**a**) Verbal fluency task (VFT) performance during NIRS sessions: comparison between thalamic stroke survivors and age-matched control subjects. Note significant word retrieval decline in the patients relative to the control group. (**b**) When patients were classified into three groups by severity of word retrieval decline, the poor and moderate groups showed remarkably low SMA responses relative to those in the healthy group. (**c**) Association of [oxy-Hb] at each region of interest (ROI) with VFT performance during NIRS. In particular, [oxy-Hb] at left and right posterior ROIs had strong correlation with word retrieval ability. ** *p* < 0.01

**Figure 5 brainsci-10-00247-f005:**
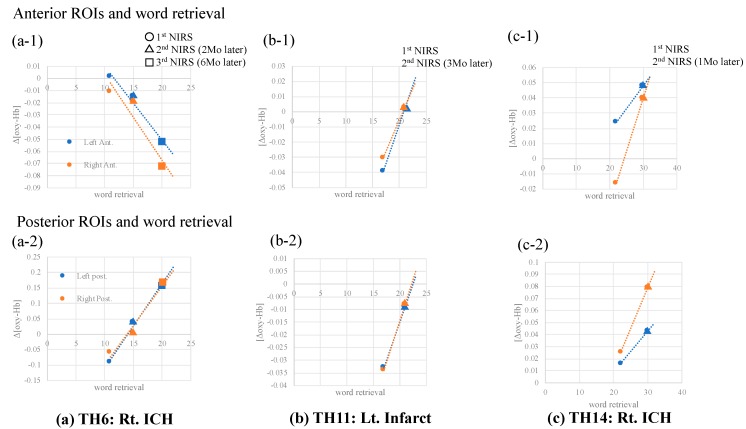
Follow-up NIRS. NIRS data were obtained from three patients ((**a**) TH6, (**b**)TH11, (**c**) TH14). All patients had improved word retrieval ability and [oxy-Hb] at almost all ROIs, together with increased VFT performance improvement. Ant.: anterior ROI; Post.:posterior ROI; Mo: month.

**Figure 6 brainsci-10-00247-f006:**
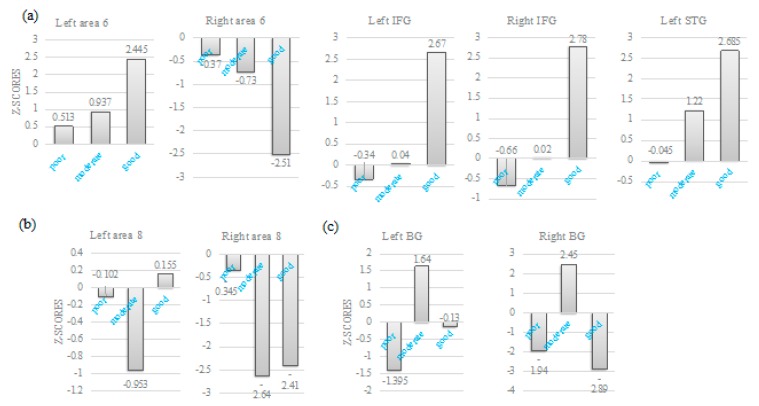
Intergroup comparison of Z-scores. (**a**) Higher Z-scores at left area 6, bilateral IFG and left superior temporal gyrus (STG) were observed with better word retrieval, while lower Z scores at right area 6 showed with better word retrieval. (**b**,**c**) Z-scores at bilateral area 8 (**b**) and bilateral basal ganglia (**c**) were seemingly independent of word retrieval disability.

**Table 1 brainsci-10-00247-t001:** Participant demographics.

	Thalamic Stroke Patients	Normal Control for f-NIRS
*n*	27	11
Age, mean (SD)	63.3 (6.17)	63.3 (7.39)
Sex (Female)	7	8
Stroke type		
Infarct	12	
Hemorrhage	15	
Laterality	Right 18 Left 7 Bilateral 1	
Cognitive impairments		
None	2	
One domain	9	
Two domains	12	
Three domains	4	
Details:		
Inattention	18	
Memory disturbance	15	
Executive dysfunction	11	
Social behavioral disorders	1	

**Table 2 brainsci-10-00247-t002:** Standard language test of aphasia (SLTA) results. Patients with dysphasia were highlighted.

Thalamic Patients	Auditory Comprehension (to Obey Verbal Commands)	Naming	Sentence Repetition	Reading Aloud Short Sentence	Reading Comprehension (to Obey Written Commands)	Dictation of Kana Letters	Dictation of Short Sentences
**TH 1**	10/10	20/20	5/5	5/5	10/10	10/10	5/5
**TH 2**	10	20	5	5	10	10	5
**TH 3**	10	20	5	5	10	10	5
**TH 4**	10	20	5	5	10	10	5
**TH 5**	10	20	5	5	10	10	5
**TH 6**	10	19	5	5	9	9	5
**TH 7**	10	20	5	5	9	-	-
**TH 8**	10	20	5	5	10	10	5
**TH 9**	10	17	4	4	5	10	2
**TH 10**	-	-	-	-	-	-	-
**TH 11**	10	19	5	5	10	10	4
**TH 12**	6	15	5	5	6	10	5
**TH 13**	9	20	5	5	10	10	5
**TH 14**	-	-	-	-	-	-	-
**TH 15**	10	20	4	5	10	10	5
**TH 16**	0	6	3	4	0	8	0

**Table 3 brainsci-10-00247-t003:** SPECT Z-scores in Brodmann areas. WNL: within normal limit, namely, scores within ± 2SD. Hypo-perfusion areas are highlighted.

SPECT Z-Scores	SMA		FAT	Frontal Cortex		Language-Related Cortical Areas			Parietal	Cerebellum	BG	
	Area 8	Area 6	IFG 44&45	MFG 46	10&11	STG 22&42	39	40	7		L	R
**TH 1**	L −2.66	L −2.5	L 2.6	L 2.44	L 2.55	L 2.22	L −2.16	L −2.82	L −2.56	L 2.35	2.35	2.47
	R −2.36	R −2.2	R 2.94	R 2.22	R 2.17	R 2.44	R 2.06	R −2.44	R 2.33	R 2.59		
**TH 2**	L 2.46	L 2.8	L −2.5	L 2.33	L −2.29	L −2.37	L 2.94	L 3.15	L 2.57	L −2.28	−2.9	−2.39
	R 2.63	R 2.61	R −2.62	R −2.16	R −2.55	R 2.64	R 2.1	R −2.40	R 2.48	R 2.32		
**TH 3**	L 2.36	L 2.81	L 3.72	L 4.0	L 2.35	L 2.35	L 2.3	L 2.63	L 2.99	R −2.55	3.12	2.16
		R −2.27	R 2.3	R 2.35	R2.2	R 2.39	R 2.18	R 2.48	R 2.72			
**TH 4**	L −2.2	L 2.36	L 2.06	normal	L -2.2	L 2.1	L 2.47	L 2.05	L -2.33	L 2.32	−2.9	−2.53
	R −2.3	R −2.38			R 2.23	R 2.12		R −2.5	R 2.17	R 2.57		
**TH 5**	L −2.4	L −2.44	L 2.15	L 2.66	L −2.15	L 2.64	L 2.74	L 2.47	R 2.45	L 2.29	−2.7	−2.41
	R −2.15	R −2.34	R 2.73	R 2.73	R −2.0	R 2.23		R −2.22		R −2.67		
**TH 6**	L 2.2	L 2.7	L 2.78	L 2.38	L −2.65	L 2.22	L 2.12	L 2.2	L 2.33	L 2.13	2.27	−2.26
	R 2.2	R 2.6	R −2.66	R −2.9	R −3		R −2.24	R 2.43	R 2.7	R 3.33		
**TH 7**	R −2.36	L 2.82	L 3.1	L 3.28	L 2.52	L 2.2	L 2.5	L −2.79	L 2.34	L 2.31	2.84	2.4
		R −2.45	R 2.68	R 2.34	R 2.32	R 2.41	R −2.41	R 2.97	R -2.7	R −2.35		
**TH 8**	L −2.22	L 2.64	L −3.4	L −2.28	L −2.89	L 2.90	L 4.39	L 2.43	L 3.05	L −2.58	4.27	2.48
	R −2.11	R 2.53	R −2.54	R −2.34	R −2.37	R 2.88	R 3.14	R 2.61	R 3.22	R −2.63		
**TH 9**	L 2.18	L 2.64	L 2.42	R −2.27	R −2.28	L 2.04	L 2.33	L −2.1	L −2.36	L −2.38		−2.58
	R 2.36	R −2.42							R 2.46	R −2.55		
**TH 10**	L 2.21	L 3.7	L −5.0	L −2.97	L −2.87	L −2.44	L −2.36	L −2.87	L 2.93	L −2.42	−2.6	−2.56
	R 2.32	R 2.93	R −3.5	R −2.89	R −2.38	R −3.28	R -2.39	R −2.28	R 2.91	R −2.42		
**TH 11**	L −3.0	L −2.64	L −2.76	L −2.5	L −2.64	L −2.59	L −2.41	L −2.60	L −2.34	L 3.73	−3.2	2.71
	R −3.18	R −2.44	R −2.62	R 2.19	R 2.2	R 2.5	R −2.58	R 2.81	R 2.52	R 3.25		
**TH 12**	L −5.22	L −5.31	L −4.87	L −4.63	L 5.47	L −2.65	L −2.66	L −3.07	L 4.38	L −3.57	−5.3	−4.83
	R −4.91	R −5.22	R −3.44	R −3.32	R 6.41	R 3.79	R 2.56	R 3.67	R 5.17	R 3.1		
**TH 13**	L 2.51	L 2.45	L −4.23	L −2.92	L −2.2	L −3.0	R 2.48	L −2.94	L 2.47	L 2.32	−2.9	−2.58
	R 2.53	R 2.45	R −2.48	R −2.65	R −2.21	R −2.50		R −2.25	R 2.09	R 2.77		
**TH 14**	L 2.51	L 2.53	L 3.28	L 2.55	L 2.80	L 3.27	L 3.63	L 2.81	L 2.81	L 3.01	2.59	−3.25
	R −2.52	R −2.65	R 2.78	R 2.78	R 2.36	R −3.14	R −2.68	R -2.59	R −2.47	R 2.79		
**TH 15**	L −2.12	L 2.51	L −2.71	L −2.36	L −2.3	L −2.31	L 2.26	L 2.8	L 2.38	L 2.36	−2.6	−2.15
	R 2.51	R 2.58		R 2.19	R −2.42	R 2.74	R 2.0	R 2.38	R 2.24	R 2.83		
**TH 16**	L −2.26	L 2.36	L −3.25	L −2.4	L −2.99	L −2.97	L −2.38	L −2.59	L −2.35	L 2.48	−2.4	2.68
	R 2.33	R 2.35	R 2.38	R 2.36	R −3.12	R 2.56	R −2.56	R 2.53	R 2.82	R 2.41		

**Table 4 brainsci-10-00247-t004:** Word retrieval performances during NIRS sessions for patients and age-matched controls. Patients were classified in terms of their word retrieval performance. Poor performance group was highlighted in orange, moderate performance in yellow, and good performance in green.

Patients	Word Retrieval during NIRS	HC	Word Retrieval
TH 1	16	HC 1	22
TH 2	6	HC 2	25
TH 3	17	HC 3	27
TH 4	26	HC 4	18
TH 5	11	HC 5	29
TH 6	11	HC 6	29
TH 7	13	HC 7	13
TH 8	16	HC 8	23
TH 9	3	HC 9	19
TH 10	1	HC 10	15
TH 11	17	HC 11	21
TH 12	13	average	21.91
TH 13	declined	SD	5.375
TH 14	22		
TH 15	10		
TH 16	0		
average	13.18		
SD	6.4

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
