# Peer review of "The Supplementary Motor Area Responsible for Word Retrieval Decline After Acute Thalamic Stroke Revealed by Coupled SPECT and Near-Infrared Spectroscopy"

_brainsci, 2020, doi:10.3390/brainsci10040247_

Round 1
Reviewer 1 Report
This manuscript presents a confirmatory analysis using SPECT and NIRS images to understand the relation between the decline in word retrieval and the thalamic stroke. The study is well-motivated and the text is clear, however, the experimental design and results need serious revisions:
1) While the main arguments and results are around the relation between SMA and word retrieval, the text in its current state fails to clearly communicate the relation between SMA and Thalamic stroke. I suggest spending some words in the introduction to clarify this issue.
2) In the last paragraph of the introduction "whether SMA activity is associated with cognitive and language function" is stated as one of the main questions to address in this study. This is while this question is already answered by same author in the previous studies (see reference [12]).
3) Excluding 3 subjects after collecting fNIRS data and just before statistical analysis is not the best design.
4) why in figure 4b, there are 13 subjects rather than 14 (in the first, third and fourth one).
5) in lines 260 and 261, it is stated that the subjects with low performance are removed from the analysis. This very selective procedure in hand-picking subjects for the final analysis is very prone to a lack of generalization in conclusions.
6) The starting points are missing in plots in Figure 5.
Overal the statistical analysis design does not seem very robust especially in subject inclusion and exclusion during the experimental procedures (Figure 1).
Author Response
Responses to Reviewer 1
1.While the main arguments and results are around the relation between SMA and word retrieval, the text in its current state fails to clearly communicate the relation between SMA and Thalamic stroke. I suggest spending some words in the introduction to clarify this issue.
Reply:
We added the following sentences (page 2, lines 51-56) to introduction section;
The ventral lateral (VL) nucleus of the thalamus projects mainly to Area 6 including SMA proper, while the mediodorsal (MD) nucleus projects to Area 8 including pre-SMA, and the medial geniculate (MG) nucleus projects to areas 41, 42 (superior temporal gyrus). Given the heterogeneity of thalamic nuclei in terms of function and projections to different areas of the cortex, it is more likely that damage to the thalamus may generate a variety of cognitive impairments.
2.In the last paragraph of the introduction "whether SMA activity is associated with cognitive and language function" is stated as one of the main questions to address in this study. This is while this question is already answered by same author in the previous studies (see reference [12]).
Reply: We changed the sentence as following;(page 3, line 73)
whether SMA activity is affected by acute phase of thalamic stroke and the changes are associated with cognitive and language function
3.Excluding 3 subjects after collecting fNIRS data and just before statistical analysis is not the best design.
Reply:
Basically, functional NIRS is sensitive to brain activity specific to a behavioral task only when subjects are devoted to performing the task. In case of very poor performance (only 0-3 words retrieved words during a total 100 seconds), the subjects seem to be silent with confusion, so their brain activity in such a condition may mainly reflect cognitive and affective components, other than word retrieval ability. In addition, excluded 3 patients could not perform even repeating a train of syllables as control task. This is why we excluded three patients from NIRS analysis. We added Table 4.
4.why in figure 4b, there are 13 subjects rather than 14 (in the first, third and fourth one).
Reply:
It is a plotting error. Figure 4b is corrected as plotted for data from 14 subjects. And data from 14 patients were used for statistical analysis.
5.in lines 260 and 261, it is stated that the subjects with low performance are removed from the analysis. This very selective procedure in hand-picking subjects for the final analysis is very prone to a lack of generalization in conclusions.
Reply:
Basically, functional NIRS is sensitive to brain activity specific to a behavioral task only when subjects are devoted to performing the task. In case of very poor performance (only 0-3 words during a total of 100 seconds in NIRS session), the subjects seem to be silent with confusion, so their brain activity in such a stressful condition may mainly contain other cognitive components due to lack of word retrieval. In addition, excluded 3 patients could not perform even repeating a train of syllables as control task. If such data were not discarded, cortical responses during NIRS session may be rather contaminated with some cognitive and affective components other than word retrieval ability. And we added Table 4.
6.The starting points are missing in plots in Figure 5.
Reply:
Additional markers (triangles and squares) were added to Figure 5.
7. Overall the statistical analysis design does not seem very robust especially in subject inclusion and exclusion during the experimental procedures (Figure 1).
Reply:
Inclusion criteria and exclusion criteria about the present study is very strict. As a result, it is a small sample size. Therefore, statistical analysis was performed by Mann-Whitney U test for nonparametric data and then showed significant results.
Reviewer 2 Report
Review for Brain Sciences
3/20/2020
This article could be very interesting to a specific group, interested in the effects of thalamic stroke.
Review Comments
Title: should remove the word ‘Coupled’. It is not clear whether the same patients had SPECT and fNIRs as indicated by the cohort chart, nor were SPECT and fNIRs actually coupled in any manner. This study was almost two separate studies.
Abstract
Should mention these are acute stroke cases. Recovery in these cases is during the acute/subacute stage, and may not reflect what has happened in chronic patients.
The study examines cognitive dysfunction…. And verbal fluency. Cognitive dysfunction is not mentioned until the end of the abstract and seems to be out of the blue.
Cohort chart
There are a few confusing items on the cohort chart.
There is a (n=2) on the left side that doesn’t seem to correspond to anything.
Also, it was confusing to see that you excluded 8 patients with aphasia. Isn’t that what you were working with? Patients with aphasia due to thalamic stroke. If not, your population needs to be better described. I don’t understand this first exclusion of 8 patients.
Please take a look at the use of wording like: On the other hand, recently and also. These often don’t need to be there and recently won’t apply years after the paper is published.
Authors cite Dick 2019, but should also read the previous paper in 2014.
Authors cite two studies suggesting damage to the SMA results in short-lived language impairment. Please do a more thorough literature search. I am not sure these are that representative. Chronic cases with SMA damage might suggest otherwise and Primary Progressive Aphasia cases might suggest otherwise.
SMA is likely important to speech production in acute, subacute and chronic phases of stroke. There is not a specific study looking at the involvement of the SMA/pre-SMA over these periods. Be careful suggesting that SMA only has an acute role.
Line 48… Increasing attention … over what time period? Last 10 years or 20 years…? This has been studied for quite a while.
Line 50… It is not clear what the focus of this study is. Is it cognitive dysfunction due to thalamic stroke or language dysfunction? It seems you are reporting two different studies here and the paper may be clearer if broken apart that way… Experiment 1. SPECT and cognitive dysfunction
Experiment 2. fNIRs and language dysfunction (verbal fluency).
Line 57: 'Given that…' This is not clear. I don’t think you meant may be similarly responsible for the loop. I think you meant that there may be a similar loop responsible for cognitive dysfunction after thalamic damage.
The last sentence in this paragraph needs some explanation. Again, this seems like it has to do with the second hypothesis or experiment in this paper- language dysfunction after thalamic stroke.
Line 61: removed ‘Coupled’ again- these are separate experiments, using different imaging/measurement techniques. They are not coupled… the patients were assigned to one or the other and did not receive both measurements… unless the patient cohort chart is not clear.
Also, you are not comparing cognitive dysfunction vs language dysfunction arising from SMA activity, but from thalamic damage. You are suggesting that SMA dysfunction plays a role…
You are measuring each one separately. – Thus item 3 in this paragraph is confusing.
Under Materials and Methods
Please separate out experiments: patients who participated in SPECT and then in a separate paragraph patients and controls participated in fNIRs study.
Again, I am confused by the cohort chart on totals and who participated in each. Did each participant complete both studies? SPECT and fNIRs? So, 16 who completed SPECT were also part of the 17 who completed fNIRs? Or did you have two groups who completed fNIRs- one set of patients who completed SPECT and then fNIRs and a separate group who only completed fNIRs…
Line 212—the frontal aslant tract connects the cortical areas 6, 8, 44 and 45. The cortical areas showed perfusion abnormalities, not the tract. The perfusion abnormalities affected many language areas. There is no causality that can be attributed here, other than thalamic stroke.
Looking at the few structural images provided… how much variance in ‘thalamic’ lesions was there. Does thalamic lesion mean that participants just had to have thalamic damage ONLY, or did they have other close regions of the striatum with damage as well.
This may have been difficult to determine very acutely…so how was this determinedl.
This could explain some of the results (remote perfusion abnormalities could be attributed to other basal ganglia regions as well.). It is hard to tell- as only 1 slice was presented. Presenting a few slices, including the lateral ventricals would be helpful.
The hyper-perfusion seen in the RH is consistent with Saur et al., 2006, 2008 – where the RH was seen to take over in the acute stage (referring to language recovery), but in the sub-acute and chronic stage, in those who had better/continued recovery, it seemed to return to the LH.
220- FAT is not a language cortical area, just a white matter pathway. Please remove and just state language relevant areas, including those connected by the FAT.
Authors should state whether SMA hyper/hypo perfusion (lines 217-218) was left or right sided in TH16 and TH12.
Authors should talk about any followup testing (3 Mo, 1 year for example) in a separate paragraph, to make it easier to follow time line of testing. 219
238 and Figure 3. It looks as though there was SMA perfusion changed in these patients. Can the authors quantify changes in left and right SMA from 1st to 2nd SPECT ? This would be very relevant to their study and is likely to correlate to their language/neuropsych scores, particularly language function.
Fix L and R labels on Figure 3 1st SPECT and double check L/R labels on 2nd SPECT
3.3.1
250 Authors have said these were thalamic aphasia cases… they should have word retrieval difficulties.., if they have aphasia.
We don’t need both a graph and text about word retrieval scores… paragraph 247 and Fig 4 are redundant.
Label the axis on Fig 4b graphs. More clearly. Vertical and horizontal.
If fNIRs data could be correlated to quantitative changes in SMA SPECT this would make this paper more interesting. They should correlate by hemisphere and ant/post.
fNIRs analysis information 3.3.2 should come before results of fNIRs 3.3
Leave out total number of participants just state 14 patients, 11 controls.
The finding that fNIRs posterior was not different between controls and patients is suprising. One would expect there would be greater LH in controls and greater RH in patients, unless all patients were very mild. Perhaps patients who were more severe and removed from analysis would show differing results from controls. Not sure why they were excluded from the analyses or not shown separately in any case. (those with very low performance). Their processing may be very relevant to the point of the study.
Again- results in 267 are redundant with graph 4b.
Graphs would be better in this instance.
If you grouped patients by severity… would you be able to see additional differences between those who were mild, moderate, severe?
Figures
What are labels a-1, b-1 and c-1? Different patients?
Again, mild patients?
Discussion around FAT pathway also needs to expand to the fact that the SMA is known to have links to the head of the caudate – was this damaged in any patients. Again, exact lesion is not fully seen with only 1 slice or fully described. SMA has multiple pathways involved in language output, not only the FAT tract. Authors should look at Naeser et al., 1989 for medial subcallosal fasciculus pathways that connect SMA through the anterior cingulate to the head of the caudate.
Authors should also look at Martin et al., 2009 Brain and Lang regarding functional activation in SMA during speech in two aphasia cases.
Not sure that involvement of both L and R in these patients is simply that they are needed for linguistic and executive function. Severity of the patients may play a role.
321- again SPECT in FAT tract is not correctly worded. SPECT in cortical regions connected by the FAT tract….
326- because the SMA has multiple connections cortical and subcortical… 326 should read … The SMA may contribute to word retrieval recovery. It could be associated with the FAT or other SMA-subcortical (basal ganglia) pathways.
- This sentence is not complete…The findings suggested the potential symptom of dysphasia after thalamic stroke.
Author Response
Response to Reviewer 2
Review Comments
1)Title: should remove the word ‘Coupled’. It is not clear whether the same patients had SPECT and fNIRs as indicated by the cohort chart, nor were SPECT and fNIRs actually coupled in any manner. This study was almost two separate studies.
Reply:
Among 16 patients undertaken SPECT, 15 patients also received NIRS measurement. 2 other patients experienced NIRS only. This is why we used “coupled”.
Abstract
2) Should mention these are acute stroke cases. Recovery in these cases is during the acute/subacute stage, and may not reflect what has happened in chronic patients.
Reply:
We added “acute” to abstract (page 1, lines 19 and 24) as well as title.
3)The study examines cognitive dysfunction…. And verbal fluency. Cognitive dysfunction is not mentioned until the end of the abstract and seems to be out of the blue.
Reply:
We added phrase like this: (Page 1 line 17)
Cohort chart
4)There are a few confusing items on the cohort chart.
There is a (n=2) on the left side that doesn’t seem to correspond to anything.
Reply:
Two patients declined to participate in SPECT, but accepted NIRS session. We added to figure 1 legend (page 4, line 99).
5)Also, it was confusing to see that you excluded 8 patients with aphasia. Isn’t that what you were working with? Patients with aphasia due to thalamic stroke. If not, your population needs to be better described. I don’t understand this first exclusion of 8 patients.
Reply:
The first excluded 8 patients was due to multiple cerebral infarctions including thalamic stroke (it is just what I meant by “multiple lesions”). Inclusion criteria included first-ever isolated thalamic infarcts or hemorrhages.
6)Please take a look at the use of wording like: On the other hand, recently and also. These often don’t need to be there and recently won’t apply years after the paper is published.
Reply:
We corrected it.
7) Authors cite Dick 2019, but should also read the previous paper in 2014. Authors cite two studies suggesting damage to the SMA results in short-lived language impairment. Please do a more thorough literature search. I am not sure these are that representative. Chronic cases with SMA damage might suggest otherwise and Primary Progressive Aphasia cases might suggest otherwise. SMA is likely important to speech production in acute, subacute and chronic phases of stroke. There is not a specific study looking at the involvement of the SMA/pre-SMA over these periods. Be careful suggesting that SMA only has an acute role.
Reply:
We cited additional papers (references 33-36). However, Dick paper in 2014 dealt with co-speech gesture, rather than language function. We focused on the involvement of SMA in early phase of recovery from language dysfunction. Therefore, we did not cite that paper.
8) Line 48… Increasing attention … over what time period? Last 10 years or 20 years…? This has been studied for quite a while.
Reply:
We changed to “So far, little attention” (page 2, line 57).
9) Line 50… It is not clear what the focus of this study is. Is it cognitive dysfunction due to thalamic stroke or language dysfunction? It seems you are reporting two different studies here and the paper may be clearer if broken apart that way… Experiment 1. SPECT and cognitive dysfunction. Experiment 2. fNIRs and language dysfunction (verbal fluency).
Line 57: 'Given that…' This is not clear. I don’t think you meant may be similarly responsible for the loop. I think you meant that there may be a similar loop responsible for cognitive dysfunction after thalamic damage.
The last sentence in this paragraph needs some explanation. Again, this seems like it has to do with the second hypothesis or experiment in this paper- language dysfunction after thalamic stroke.
Reply:
We changed the description as following (page 3, line 65):
The focus of the present study is on both of cognitive dysfunction and language dysfunction due to thalamic stroke. We propose two hypotheses. One is that cognitive impairment due to acute thalamic stroke may be related to the fronto-brainstem-cerebellar-thalamic loop, as is similar with the underlying mechanism of cognitive dysfunction due to acute pontine stroke. The other is that language dysfunction due to acute thalamic stroke may be responsible for the language-related brain regions including the SMA, IFG, STG, angular gyrus and supramarginal gyrus.
10) Line 61: removed ‘Coupled’ again- these are separate experiments, using different imaging/measurement techniques. They are not coupled… the patients were assigned to one or the other and did not receive both measurements… unless the patient cohort chart is not clear.
Reply:
Among 16 patients undertaken SPECT, 15 patients also received NIRS measurement. 2 other patients experienced NIRS only. This is why we used “coupled SPECT and NIRS”.
11)Also, you are not comparing cognitive dysfunction vs language dysfunction arising from SMA activity, but from thalamic damage. You are suggesting that SMA dysfunction plays a role…
You are measuring each one separately. – Thus item 3 in this paragraph is confusing.
Reply:
We tested two hypotheses in the present study. One is that cognitive impairment due to acute thalamic stroke may be related to the fronto-brainstem-cerebellar-thalamic loop, as is similar with the underlying mechanism of cognitive dysfunction due to acute pontine stroke. The other is that language dysfunction due to acute thalamic stroke may be responsible for the language-related brain regions including the SMA, IFG, STG, angular gyrus and supramarginal gyrus.
Under Materials and Methods
12)Please separate out experiments: patients who participated in SPECT and then in a separate paragraph patients and controls participated in fNIRs study. Again, I am confused by the cohort chart on totals and who participated in each. Did each participant complete both studies? SPECT and fNIRs? So, 16 who completed SPECT were also part of the 17 who completed fNIRs? Or did you have two groups who completed fNIRs- one set of patients who completed SPECT and then fNIRs and a separate group who only completed fNIRs…
Reply:
Among 16 patients undertaken SPECT, 15 patients also received NIRS measurement. 2 other patients experienced NIRS only.
13)Line 212—the frontal aslant tract connects the cortical areas 6, 8, 44 and 45. The cortical areas showed perfusion abnormalities, not the tract.
Reply:
We changed as following: perfusion abnormality involved bilateral SMA and IFG connected by the FAT pathway (Page 9, lines 223, and 226 in revised version).
14)The perfusion abnormalities affected many language areas. There is no causality that can be attributed here, other than thalamic stroke. Looking at the few structural images provided… how much variance in ‘thalamic’ lesions was there. Does thalamic lesion mean that participants just had to have thalamic damage ONLY, or did they have other close regions of the striatum with damage as well. This may have been difficult to determine very acutely…so how was this determinedl. This could explain some of the results (remote perfusion abnormalities could be attributed to other basal ganglia regions as well.). It is hard to tell- as only 1 slice was presented. Presenting a few slices, including the lateral ventricals would be helpful.
Reply:
Inclusion criteria for neuroimaging included patients under age 80, with first-ever isolated thalamic infarcts or hemorrhages without motor deficits. Exclusion criteria were a past or current history of psychiatric or neurological illnesses as well as those with motor impairment due to the ictus. If the thalamic damage extends to the striatum through internal capsule, motor deficits are inevitable. Such a patient with motor deficits was excluded. Accordingly, many language-related cortical perfusion abnormalities may be due to acute phase of first-ever isolated thalamic stroke.
15)The hyper-perfusion seen in the RH is consistent with Saur et al., 2006, 2008 – where the RH was seen to take over in the acute stage (referring to language recovery), but in the sub-acute and chronic stage, in those who had better/continued recovery, it seemed to return to the LH.
Reply:
We have argued this point in page 16 line 360 in revised version.
16)220- FAT is not a language cortical area, just a white matter pathway. Please remove and just state language relevant areas, including those connected by the FAT.
Reply:
We corrected it.
17)Authors should state whether SMA hyper/hypo perfusion (lines 217-218) was left or right sided in TH16 and TH12.
Reply: (page 9 lines 228-230 in revised version)
With respect to thalamic aphasia, two patients (TH12 and 16) presenting with global aphasia, commonly yielded hypo-perfusion in Brodmann areas 8, 44 & 45 connected by the FAT, and other language-related brain areas in the left hemisphere (22 & 42, 39, 40) and hyper-perfusion in right language homologous areas (22 & 42, 40) (Figure 2-b,c). One patient (TH16) revealed hyper-perfusion at bilateral SMA-proper (area 6) while the other (TH12) showed bilateral hypo-perfusion.
18)Authors should talk about any followup testing (3 Mo, 1 year for example) in a separate paragraph, to make it easier to follow time line of testing. 219
Reply:
We corrected (page 9, line 230).
19)238 and Figure 3. It looks as though there was SMA perfusion changed in these patients. Can the authors quantify changes in left and right SMA from 1st to 2nd SPECT ? This would be very relevant to their study and is likely to correlate to their language/neuropsych scores, particularly language function.
Reply:
We quantified perfusion changes in SMA of a patient (TH6) by Z scores as shown in figure 3-d. In particular, left 8 area remarkably changed 6 Mo later: hyper-perfusion into hypo-perfusion, associated with improved word retrieval ability. In addition, we showed the long-term relationship between [oxy-Hb] changes in SMA and word retrieval as figure 5-a. As shown in Figure 5 a-1, we demonstrated negative correlation between [oxy-Hb] changes in anterior ROI of SMA and word retrieval. NIRS dynamic responses may be presumably influenced by change of hyper-perfusion at SMA into hypo-perfusion.
20)Fix L and R labels on Figure 3 1st SPECT and double check L/R labels on 2nd SPECT
Reply:
We corrected labels.
3.3.1
21)250 Authors have said these were thalamic aphasia cases… they should have word retrieval difficulties.., if they have aphasia. We don’t need both a graph and text about word retrieval scores… paragraph 247 and Fig 4 are redundant.
Reply:
We edited this paragraph as following: (page 11, line 264 in revised version)
The findings suggested that thalamic stroke survivors were characterized by…, regardless of absence of aphasia as shown by Table 2.
22)Label the axis on Fig 4b graphs. More clearly. Vertical and horizontal.
Reply:
We corrected Figure 4c.
23)If fNIRs data could be correlated to quantitative changes in SMA SPECT this would make this paper more interesting. They should correlate by hemisphere and ant/post.
Reply:
There was no significant correlation between SMA SPECT Z-scores and [oxy-Hb] from NIRS.
24)fNIRs analysis information 3.3.2 should come before results of fNIRs 3.3
Reply:
We changed 3.3.2. subheading as “NIRS results: the comparison with the control and the correlation” (page 12, line 279 in revised version).
25)Leave out total number of participants just state 14 patients, 11 controls.
Reply:
We corrected plots in figure 4c.
26)The finding that fNIRs posterior was not different between controls and patients is suprising. One would expect there would be greater LH in controls and greater RH in patients, unless all patients were very mild. Perhaps patients who were more severe and removed from analysis would show differing results from controls. Not sure why they were excluded from the analyses or not shown separately in any case. (those with very low performance). Their processing may be very relevant to the point of the study.
Reply:
There were no significant differences at either ROIs between the two groups. We demonstrated that none of thalamic stroke survivors analyzed had dysphasia as shown in Tables 2 and 4. Therefore, SMA responses from NIRS during VFT in such a condition would reflect other than dysphasia. Word retrieval ability requires executive function as well as language one, so the decline in thalamic stroke group may involve both executive dysfunction and language one. in addition, excluded 3 patients could not perform even repeating a train of syllables as control task. We could not find any physiological significance of SMA responses during no utterance in NIRS session.
27)Again- results in 267 are redundant with graph 4b.
Graphs would be better in this instance.
Reply:
The description of 3.3.2. NIRS results (page12 line 271) as well as Figure 4c is useful for readers to understand this study.
28)If you grouped patients by severity… would you be able to see additional differences between those who were mild, moderate, severe?
When we grouped thalamic patients by severity of word retrieval difficulty (poor, moderate and good performance), we could find additional differences among groups. We added figures (Figures 4b and 6) and the explanation. And then we changed label of Figure 4b to 4c.
Page 13, line 287: When patients were classified into 3 groups by severity of word retrieval decline, NIRS data showed a trend toward the association of [oxy-Hb] and word retrieval ability (Figure 4b). Especially, poor and moderate group showed remarkable low SMA responses relative to those with healthy group.
Page 14, line 312:
3.3.4. NIRS data and SPECT Z-scores
When patients were classified into 3 groups by severity of word retrieval decline, some of language-related brain regions were sensitive to the classification. Specifically, higher Z-scores at left posterior SMA, IFG and left STG were observed with word retrieval performance was better while lower scores at right posterior SMA showing better performance (Figure 6). On the contrary, Z-scores at bilateral area 8 (b) and bilateral Basal ganglia (c) were seemingly independent of word retrieval disability.
Figures
29)What are labels a-1, b-1 and c-1? Different patients?
Again, mild patients?
Reply:
In Figure 5, we expressed data from three different patients (assigned TH 6, 11, and 14). TH6 (11 words at the 1st NIRS) belongs to poor group, TH 11 (17 words) to moderate, and TH 14 (22 words) to good group. At final NIRS session, all patients improved the word retrieval ability beyond the average of health control group.
30)Discussion around FAT pathway also needs to expand to the fact that the SMA is known to have links to the head of the caudate – was this damaged in any patients. Again, exact lesion is not fully seen with only 1 slice or fully described.
Reply:
We confirmed that all of the enrolled patients had first-ever isolated thalamic stroke by careful checking multi-slice and various modes of MRI images, based on our Inclusion criteria and exclusion criteria. However, checking whether damage to thalamus would affect the basal ganglia or not, we found another perfusion abnormality in bilateral basal ganglia. So, we changed table 3 (page 10, line 235) and added Figure 6-c (page 14, line 318).
31) SMA has multiple pathways involved in language output, not only the FAT tract. Authors should look at Naeser et al., 1989 for medial subcallosal fasciculus pathways that connect SMA through the anterior cingulate to the head of the caudate.
Reply: page 17, line 387
Also, our SPECT results showed perfusion abnormality in bilateral basal ganglia, suggesting another association with word retrieval difficulty after acute thalamic stroke. The view is explained by the fact that SMA also connects with the basal ganglia [33]. Earlier papers argued the contribution of basal ganglia and SMA to recovery from aphasia [34, 35].
32)Authors should also look at Martin et al., 2009 Brain and Lang regarding functional activation in SMA during speech in two aphasia cases.
Not sure that involvement of both L and R in these patients is simply that they are needed for linguistic and executive function. Severity of the patients may play a role.
Reply:
As Martin et al suggested, we confirmed that the SMA function during word retrieval was different between left SMA and right, as is shown in Figure 6. Different point is that bilateral SMA may play a different role on word retrieval ability. We added this to page 16 line 360.
An earlier study suggested that left SMA may be involved in the improvement of aphasia by rTMS [26]. Similarly, our results suggested that left SMA showed positive correlation while right SMA showing negative correlation. The role of SMA on word retrieval ability might be the laterality-dependent and at the very least, left and right SMA may play a crucial role on word retrieval in different way.
33)321- again SPECT in FAT tract is not correctly worded. SPECT in cortical regions connected by the FAT tract….
Reply:
We corrected it.
34)326- because the SMA has multiple connections cortical and subcortical… 326 should read … The SMA may contribute to word retrieval recovery. It could be associated with the FAT or other SMA-subcortical (basal ganglia) pathways.
Reply:
We added the following in Discussion (page 17, line 387).
Also, our SPECT results showed perfusion abnormality in bilateral basal ganglia, suggesting another association with word retrieval difficulty after acute thalamic stroke. The view is explained by the fact that SMA also connects with the basal ganglia [33]. Earlier papers argued the contribution of basal ganglia and SMA to recovery from aphasia [34, 35].
35)333.This sentence is not complete…The findings suggested the potential symptom of dysphasia after thalamic stroke.
Reply: page 17, line 376
We changed it as following:
The findings may include the compensate mechanism to protect against dysphasia after thalamic stroke.
Round 2
Reviewer 1 Report
Thank you for the author's clarifications and revisions. The revised text is more clear and accurate. However, my concern regarding the research design stands as before. The author's argument about the sensitivity of fNIRS to functional brain activity and possible contamination of recorded data with other cognitive components is general for all functional neuroimaging modalities (is not specific to fNIRS).
Author Response
- Excluding 3 subjects after collecting fNIRS data and just before statistical analysis is not the best design.
- However, my concern regarding the research design stands as before. The author's argument about the sensitivity of fNIRS to functional brain activity and possible contamination of recorded data with other cognitive components is general for all functional neuroimaging modalities (is not specific to fNIRS).
Reply:
Given that non-aphasic patients had good performance of naming (see Table 2), their word retrieval difficulty could not be due mainly to dysphasia. Phonemic VFT requiring speech production involves both language function and executive one and it is difficult to isolate each other. At least, word retrieval of the aphasic patients was more affected by aphasia than executive function whereas those of the non-aphasic patients affected more by executive function. As mentioned in introduction section, we used SPECT and NIRS for different purposes. We used SPECT to detect the mechanism underlying the cognition and language function affected by thalamic stroke. So, SPECT scan in rest condition was available for even patients with aphasia. As a result, the perfusion abnormality included some cognitive functions such as aphasia, word retrieval and dysexecutive function. On the contrary, we used NIRS to exclusively focus on the role of SMA on word retrieval disability after thalamic stroke, rather than thalamic aphasia. In addition, VFT task used in NIRS session was not suitable for aphasic patients. Because both of VFT task and control task require verbal response and the successful NIRS measurement is guaranteed by the quality of task performance. We collected NIRS data from 16 patients including aphasic ones. As expected, aphasic patients had very poor performance (only 0-3 words retrieved words during a total 100 seconds). In addition, they could not repeat a train of syllables as control task thoroughly.Throughout the NIRS session, they seemed to be silent with confusion. Therefore, their brain activity in such a condition may mainly reflect cognitive and affective components, other than word retrieval ability. Therefore, we excluded aphasic patients from NIRS analysis.
Page 13, lines 366-368: bellow sentences were deleted according to above discussion.
The idea may be supported by our NIRS data showing a close association of SFX hemodynamic responses during phonemic VFT with word retrieval ability.
Instead, an augment was added at page 13, line 348 as following:
Given that non-aphasic patients had good performance of naming (see Table 2), their word retrieval difficulty could not be due mainly to dysphasia. Phonemic VFT requiring speech production involves both language function and executive one. Although it is difficult to isolate each other, at least, word retrieval of aphasic patients was more affected by aphasia than executive function whereas those of non-aphasic patients was more affected by executive function. We collected NIRS data from 16 patients including aphasic ones. As expected, aphasic patients had very poor performance (only 0-3 words retrieved words during a total 100 seconds). In addition, they could not repeat a train of syllables as control task thoroughly. Throughout the NIRS session, they seemed to be silent with confusion. As a result, their brain activity in such a condition may mainly reflect cognitive and affective components, other than word retrieval ability. Therefore, we excluded aphasic patients from NIRS analysis.
Reviewer 2 Report
Please see minor comments. - these would just clarify some areas a bit.
Thank you for your responses, the paper reads much better.

Author Response
- page 1, line 27: which side? → Reply: bilateral SMA
- page 1, line 31 → Reply: added “ after thalamic stroke”
- page 2, line 43 → Reply: change “The supplementary ….”
- page 2, line 47: → Reply: change “some” cases,
- page 2, line 55: → Reply: added “ mostly related to language processing” here
- page 2, line 56: → Reply: changed “it is also” here
- page 8, line 231: → Reply: deleted “The”
- page 10, line 241: →Reply: added “ initial” here
- page 11, line 251: Did this include hyperperfusion of L and R SMA? Did L and R SMA perfusion change after 2nd SPECT?
Reply: page 11, line 253: And right SMA consistently showed hyper-perfusion whereas left SMA perfusion changed in a reversed manner (Figure 3-d).
- page 13, line 297-299: added SMA there
- page 13, line 303: added SMA here